# Ultrasound-Guided Injection of Autologous Platelet-Rich Plasma for Refractory Lateral Epicondylitis of Humerus: Case Series

**DOI:** 10.3390/jpm13010066

**Published:** 2022-12-28

**Authors:** Guohang Huang, Jiangshan Zhang, Zhenhai Wei, Yiying Mai, Jisheng Guo, Li Jiang

**Affiliations:** 1The Third Affiliated Hospital, Sun Yat-sen University, Guangzhou 510275, China; 2The First School of Clinical Medicine, Southern Medical University, Guangzhou 510080, China; 3The Sixth Affiliated Hospital, Sun Yat-sen University, Guangzhou 510275, China

**Keywords:** platelet-rich plasma, refractory lateral epicondylitis, musculoskeletal ultrasonography, case report

## Abstract

Refractory lateral epicondylitis (RLE) is a tendinopathy of the elbow with less effective conservation treatment. Platelet-rich plasma (PRP) is a new treatment option for RLE because of its repair-promoting effect on tissues. Although evidence demonstrates the efficacy of PRP in treating tendinopathies, the therapeutic utility of ultrasound-guided PRP injection for RLE is unknown. Here, we report two cases of RLE treated with PRP. The first patient was a 78-year-old man who received an unknown dose of local glucocorticoid injection at the local community clinic in June 2016. His pain recurred after exertion. The second patient was a 54-year-old woman who received a glucocorticoid injection (0.5 mL of compound betamethasone and 1.5 mL of 0.9% normal saline) in October 2020. Her pain could not be relieved. A physician diagnosed patients with RLE based on their medical history, symptoms, and clinical signs. The doctor injected PRP (the first patient in November 2020, the second in March 2021) under ultrasound guidance into the patient’s attachment point of the extensor tendon at the lateral humeral epicondyle. The doctor evaluated the effectiveness of the treatment by ultrasonography, visual analogue scale, and the patient-rated tennis elbow evaluation. After four weeks of treatment, the pain was relieved, and functions continued to improve in the following three months. Moreover, the ultrasonography showed that the damaged tendons were repaired. Together, we demonstrate that ultrasound-guided PRP injection could considerably relieve pain, improve elbow joint functions in patients with RLE, and provide visible evidence that PRP repairs tendon damage.

## 1. Introduction

Lateral epicondylitis, or tennis elbow, is a tendinopathy caused by acute or chronic injury. The prevalence of lateral epicondylitis is 1% to 3% in 35- to 55-year-olds, especially in persons who overuse or overload the wrist extensor muscles [1,2]. The characteristic pathological changes of the condition are the tearing of the extensor tendon attached to the lateral epicondyle of the humerus, accompanied by the proliferation of vascular fibroblasts [3]. Patients with lateral epicondylitis usually complain of pain spreading from the lateral side of the elbow to the forearm, affecting most activities of daily living. In severe cases, the pain may last for several weeks or months, causing tremendous economic and psychological burdens. If the conservative treatment proves unsuccessful after at least six months, the patient is diagnosed with refractory lateral epicondylitis (RLE).

Common conservative treatment methods for lateral epicondylitis encompass oral non-steroidal anti-inflammatory drugs, physical therapy, and steroid injections [4]. Their drawbacks are variable effectiveness, low tissue-restoring ability, and lack of imaging evidence of structural improvement. Approximately 3% to 11% of patients with the condition eventually require operative intervention [5], especially those with RLE. Since many attractive management strategies remain controversial [4], researchers aim to reduce the operation rate by improving conservative treatment methods. Platelet-rich plasma (PRP) is abundant in platelets, growth factors, and fibrin. It promotes the proliferation of tendon cells and angiogenesis, and has an anti-inflammatory and tissue-repairing potential [6]. Recent evidence suggests PRP reduces the need for surgical intervention and represents an alternative treatment option for patients with RLE who refuse to undergo surgical operation [7]. Furthermore, a study with an average follow-up time of 5.2 years showed that PRP substantially improves the symptoms of patients with RLE and reduces the incidence of surgical procedures [8]. In addition, PRP injection has an advantage over corticosteroid injection since it improves the functional activity score and reduces pain and the risk of complications during long-term follow-up [9,10]. Although PRP seems a promising treatment strategy, some researchers question whether it is effective in treating chronic lateral epicondylitis [11], since conclusions may differ when drawn from different preparation methods and treatment regimens.

Since controversy remains about the effectiveness of PRP in treating RLE and lack of direct evidence obscures the repair effect of PRP on muscles and tendons, we report two patients with RLE whom we treated with ultrasound-guided PRP in our clinic. We routinely use ultrasound-guided injections for musculoskeletal pain in the clinic and demonstrate that they aid in administering PRP to relieve pain and improve function. In addition, we show that ultrasound images taken before and after PRP injection prove it repairs injured tendons. This case report investigates the effect of PRP on RLE and whether musculoskeletal ultrasound can be used to visualize PRP-mediated repair of damaged tendon tissue.

## 2. Materials and Methods

### 2.1. Patient Information

Two patients with RLE were admitted to the Rehabilitation Department of the Third Affiliated Hospital of Sun Yat-sen University. The following case description adheres to the CARE (case report) guidelines [12].

#### 2.1.1. Case One Description

A 78-year-old man with more than four years’ history of intermittent pain on the lateral side of the left elbow presented with aggravated pain intensity lasting four months. In June 2016, he was locally injected with an unspecified dose of glucocorticoids at the local community clinic, and the pain disappeared. However, it recurred after exertion, and the patient used a topical anti-inflammatory analgesic patch to relieve the pain. In July 2020, the patient experienced pain in his left elbow again after lifting a heavy object, which manifested as persistent pain. When it became severe, the patient’s activities of daily living, such as holding items or folding towels, were restricted, and the topical patch was ineffective. The patient went to the local clinic again and requested the local injection treatment, but did not receive it since it did not improve his condition. He eventually arrived at our outpatient clinic for further treatment in November 2020. After undergoing physical and ultrasound examinations, the patient received an ultrasound-guided injection of PRP.

##### Physical Examination

While no obvious redness and swelling were found on the lateral side of the patient’s left elbow, marked tenderness of the lateral epicondyle of the humerus was observed. The active and passive flexion and extension activities of the left elbow joint were slightly restricted, and the rotation movement of the forearm was severely restricted. Visual analogue scale (VAS) scores were 4/10 at rest and 7/10 with movement. Patient-rated tennis elbow evaluation (PRTEE) was 53.5 points/100 points (pain subscale score 27/50; functional subscale score 26.5/50).

##### Characteristic of Ultrasonography

The extensor tendon attached to the lateral epicondyle of the left humerus was torn. The tendon tear was characterized by anechoic fiber disruption. The tendon was thick, and the surface of the cortical bone was rough at the attachment point of the extensor tendon. This case is shown in Figure 1.

#### 2.1.2. Case Two Description

A 54-year-old woman presented with seven months of pain on the lateral side of the right elbow. The pain interfered with daily work, such as sweeping the floor, cooking, and lifting light-weight objects, and caused sleep disruption, seriously affecting the patient’s quality of life. She was treated in another hospital in October 2020 with a local hormone injection containing 0.5 mL of compound betamethasone mixed with 1.5 mL of 0.9% normal saline. Her symptoms did not improve after the treatment. In March 2021, she visited our outpatient clinic, where she underwent physical and ultrasound examinations and received an ultrasound-guided injection of PRP.

##### Physical Examination

Severe tenderness was noted on the lateral side of the patient’s right elbow without obvious redness and swelling. Finger extension and wrist extension caused pain. The passive flexion and extension activities of the right elbow joint were slightly restricted, whereas the rotation movement of the forearm was severely restricted. Passive wrist flexion test (Mill’s test) was positive, while upper limb neural tension test was negative. Visual analogue scale scores were 4/10 at rest and 8/10 with movement. Patient-rated tennis elbow evaluation was 65.5 points/100 points (pain subscale score 30/50; functional subscale score 35.5/50).

##### Characteristic of Ultrasonography

The extensor tendon attached to the lateral epicondyle of the right humerus was torn. The tendon was noticeably swollen, with hypoechoic or anechoic lesions. This case is depicted in Figure 2.

### 2.2. Diagnostic Assessment

Initial ultrasound evaluation and pain and function assessment were performed before the PRP treatment (baseline) and four weeks or three months from the baseline. In case two, an additional ultrasound verification was done two weeks from the baseline. Both patients were followed up for pain and function scores three months after the first PRP injection.

Ultrasonography: The patient’s elbow joint was flexed at 90°. The forearm was internally rotated, with the palm down. The ultrasound probe was parallel to the radius, placed between the radial head and the lateral epicondyle of the humerus. This position allows the radial head, the anterolateral surface of the humerus, and the surrounding soft tissues to be displayed on the ultrasound system. It also permits the visualization of the common extensor tendon and the radial collateral ligament attached to the lateral epicondyle.

Visual analogue scale (VAS): Numbers from 0 to 10 indicated the degree of pain (0 = no pain, 10 = extreme pain). Patients were asked to choose a number that most faithfully describes the experienced pain. The higher the number, the more severe the pain.

Patient-Rated Tennis Elbow Evaluation (PRTEE): The PRTEE questionnaire is an reliable, reproducible assessment of chronic elbow tendinopathy [13]. The patients rated the levels of their elbow pain and disability using two subscales: pain subscale (0 = no pain, 10 = extreme pain) and function subscale (0 = can do without difficulty, 10 = unable to do). In addition to the individual subscale scores, a total score was computed on a 0 to 100 scale (0 = no disability), where pain and functional problems were weighted equally.

### 2.3. Therapeutic Intervention

#### 2.3.1. Preparation of PRP

The procedure was performed as follows: A total of 20 mL of the patient’s autologous whole blood was evenly distributed into two 10-mL EDTA (ethylenediaminetetraacetic acid)-coated tubes. The platelet concentration was determined, and the tubes were centrifuged at 500× *g* for 8 min.The upper layer (blood plasma) was pipetted into a sterile tube and mixed. The platelet concentration (a) was assessed in the pipetted plasma volume (b), followed by centrifugation at 1900× *g* for 12 min.The upper layer (platelet-poor plasma with an approximate volume of a × b/1000) was discarded, leaving a thin layer of plasma above the platelet pellet. The pellet was resuspended with a sterile pipette, and the platelet concentration was calculated (approximately 1000 × 10^9^/L). A syringe was used to draw 3 mL of PRP, which was mixed with 0.5 mL of lidocaine for subsequent injection. The PRP preparation should be completed within 30 min after the blood draw to preserve the activity of PRP. Likewise, PRP should be administered immediately after the preparation. ([a] and [b] represent the platelet concentration and plasma volume in the upper layer obtained after the first centrifugation).

#### 2.3.2. Ultrasound-Guided Injection Technique

An aseptic technique was used during the PRP injection. It was performed under ultrasound guidance using a Sonosite S Series ultrasound device (FUJIFILM Sonosite Inc., Bothell, WA, USA) and an HFL38x high-frequency linear array ultrasound probe transducer (13–6 MHz, ultra-wide frequency conversion). The patient’s elbow joint was flexed at 90°, with the forearm internally rotated and the palm down. The ultrasound probe was parallel to the radius, placed between the radial head and the lateral epicondyle of the humerus. Once per week for four weeks, 3 mL of PRP was injected into the attachment point of the common extensor tendon at the lateral epicondyle of the humerus.

The PRP preparation was done by experienced technicians at the Third Affiliated Hospital of Sun Yat-sen University. In addition, PRP should be administered immediately once prepared. The rehabilitation physician at our hospital injected PRP within 5 to 10 min following the preparation. The ultrasound-guided PRP injection is explained in Figure 3.

## 3. Results

### 3.1. Pain and Functional Outcome

#### 3.1.1. Case One

The elbow joint function and the pain score improved considerably after the PRP treatment versus the baseline (Table 1). Four weeks from the baseline, the visual analogue scale (VAS) values for pain during rest and activity decreased by 50% and 57.1%, respectively. Remarkably, three months from baseline, the VAS values dropped further by 100% and 71.4%, respectively. The patient-rated tennis elbow evaluation (PRTEE) score declined by 71.0% (pain subscale 66.7%; functional subscale 75.5%) four weeks from the baseline, and by 87.9% (pain subscale 85.2%; functional subscale 90.6%) after three months.

The patient’s activities of daily living involving wrists and hands, such as lifting and folding objects, were obviously restricted before the treatment. Four weeks from the baseline, the patient recuperated the ability to do the activities and felt soreness on the outer side of the elbow only while doing something laborious. Three months from the baseline, the patient was pain free, only occasionally experiencing discomfort on the outer side of the elbow while folding items or unfastening them by a turning motion. Data are summarized in Table 1.

#### 3.1.2. Case Two

The elbow joint function and the pain score improved rapidly following the PRP treatment (Table 2). Four weeks from the baseline, the VAS values at rest and activity decreased by 50%. Three months from the baseline, these values declined further by 100% and 75%, respectively. Similarly, the PRTEE score decreased by 72.5% (pain subscale 66.7%; functional subscale 77.5%) four weeks after the baseline, and by 86.3% (pain subscale 83%, functional subscale 88.7%) after three months.

Before the treatment, the patient was unable to do various activities of daily living, such as sweeping, cooking, and picking up light objects, due to pain. Two weeks from the baseline, the patient could already perform some activities but could not lift light objects. At three weeks, she regained the ability to lift light objects. At four weeks, she could lift heavy objects but with moderate difficulty. Three months from the baseline, the patient could fold and unfasten objects and pick up heavy items with only mild discomfort. Data related to this case are shown in Table 2.

### 3.2. Ultrasonography Evaluation

Both patients underwent ultrasonography evaluation at baseline and four weeks following it. Detailed ultrasound images are depicted in Figure 4 (case one) and Figure 5 (case two). An additional ultrasonography assessment was done two weeks after the baseline for case two to visualize tissue growth during the PRP treatment (Figure 5).

#### 3.2.1. Case One

Four weeks after the baseline, the ultrasound images showed that the swelling of the damaged tendon substantially subsided. The thickening of the extensor tendon attached to the lateral epicondyle of the humerus was 0.49 cm at baseline but reduced to 0.40 cm four weeks from the baseline. The anechoic area formed by the tendon tear shrank noticeably. In addition, the morphology and integrity of the tendon resembled that of the unaffected side. However, the cortical bone at the attachment point of the total extensor tendon retained a rough appearance (Figure 4).

#### 3.2.2. Case Two

Two and four weeks after the baseline, the ultrasound images revealed that the swelling of the damaged tendon gradually subsided. The thickening of the extensor tendon attached to the lateral epicondyle of the humerus was 0.70 cm at baseline but decreased to 0.55 cm at two weeks and 0.41 cm at four weeks. The anechoic area formed by the extensor tendon tear was reduced continuously, but the cortical bone at the attachment point of the total extensor tendon remained rough (Figure 5).

### 3.3. Complications and Adverse Events

No serious adverse events occurred during the entire treatment (one PRP administration per week for four weeks). After receiving the first PRP injection, only one patient experienced pain at the injection site. Nonetheless, after each consecutive injection, the degree of pain reduced, and its duration gradually shortened.

## 4. Discussion

Most patients with lateral epicondylitis can be cured with conservative treatment approaches. However, for those with refractory lateral epicondylitis (RLE), conservative treatment is usually less effective [4]. Regrettably, from 3% to 11% of these patients eventually require a surgical intervention [5], which many may aim to avoid due to high costs and developed anxiety. Moreover, approximately 9% of patients continue having moderate to severe pain, and up to 28% experience mild pain five years after a surgical procedure [14]. Platelet-rich plasma (PRP) is a biological therapy that may promote tissue repair. It provides an alternative option for patients unwilling to undergo an operation or whose conservative treatment has failed. It is abundant in diverse cytokines that change the molecular microenvironment of the injured site and activate the healing mechanism. Thus, PRP promotes inflammation regression, proliferation, and stem-cell differentiation into tenocytes [15]. Patients with RLE treated with PRP experience faster improvement in limb functions, pain scores, and mood than other treatments (e.g., corticosteroids, bupivacaine, autologous blood, placebo, etc.) in the long-term follow-up [16,17,18].

A recent meta-analysis challenges the effectiveness of PRP in treating chronic lateral epicondylitis since the treatment offers no significant improvement compared with others [19]. However, because this study compares few publications and has a small effect size, its conclusions about PRP are questionable. We verified six publications from the meta-analysis and discovered that the PRP composition and dose vary between the studies [20]. Moreover, two of them do not report platelet concentrations [18,21]. These findings suggest reasons for the elusive conclusions about whether PRP is an effective treatment strategy for RLE. Another recent meta-analysis supports our results. It shows that during short- or long-term follow-up, the pain scores of patients with lateral epicondylitis are significantly lower than those in the control group. In addition, the PRP injection does not cause any obvious adverse reaction [22]. Similarly, PRP treatment significantly improves pain and function in patients with RLE versus healthy persons [17]. The above evidence agrees with the results observed in this case report. After receiving PRP injection, pain and function in two patients with RLE have continuously and considerably improved during a three-month follow-up, demonstrating the beneficial effect of this approach on musculoskeletal tissues.

Most studies using PRP as a treatment for RLE focus on functional improvement and pain reduction, without providing direct evidence of tissue repair [23]. Ultrasound evaluation of the lateral epicondyle of the humerus is an effective, noninvasive, and relatively cheap method. It has high specificity and sensitivity (approximately 80%) in distinguishing the histological changes associated with epicondylitis [24]. These include thickened and swollen tendons, tears at the tendon attachment points, structural disorders, calcifications, and gaps in the cortical bone. The method can also find new blood vessels if combined with the color doppler. In this study, we used musculoskeletal ultrasound in the auxiliary diagnosis of RLE, guiding the PRP injection therapy, and in the follow-up observation after the treatment. We showed that the swelling of the common extensor tendon completely subsided following PRP treatment. The structure of the tendon was clear, suggesting a restored morphology and integrity of the tendon. Thus, ultrasound evaluation provides visual evidence for PRP-promoted repair and regeneration of damaged tendons. We believe that using ultrasound diagnosis and treatment technology for evaluating RLE and guiding precision injection therapy should be encouraged in a clinical setting.

If only tendon injury evaluation is required, ultrasound is generally considered the first line diagnostic tool. If the patient has injuries other than tendinopathy, plain film or magnetic resonance may be considered for further evaluation of other structural injuries [25]. In this study, musculoskeletal ultrasound clearly showed the image characteristics of the patient’s tendon tissue before and after treatment, providing an objective image basis for the curative effect.

In summary, the clinical results of this case report support PRP as a treatment for RLE and provide direct evidence that PRP repairs damaged tendon tissue. In addition, they suggest that ultrasound image-guided technology can accurately inject PRP into the injured site, which is the cornerstone of successful treatment. We propose this study as an objective basis for RLE treatment and management.

## 5. Conclusions

We have reported two cases of RLE patients who received ultrasound-guided injection of PRP. As a result, the pain symptoms of the patients were significantly relieved or even eliminated, the function was basically restored, and there were no adverse events. Ultrasound imaging showed that the patient’s tendon swelling was significantly reduced in a short period of time after PRP treatment, and the morphology and integrity of the tendon were significantly improved, even close to the shape of the healthy side. This case report indicates that the treatment strategy of PRP injection may help repair damaged tendons in RLE patients. Ultrasound imaging technology provides a reliable tool for precise injection and assessment of tendon changes, and is worthy of being widely promoted in clinical applications.

## Figures and Tables

**Figure 1 jpm-13-00066-f001:**
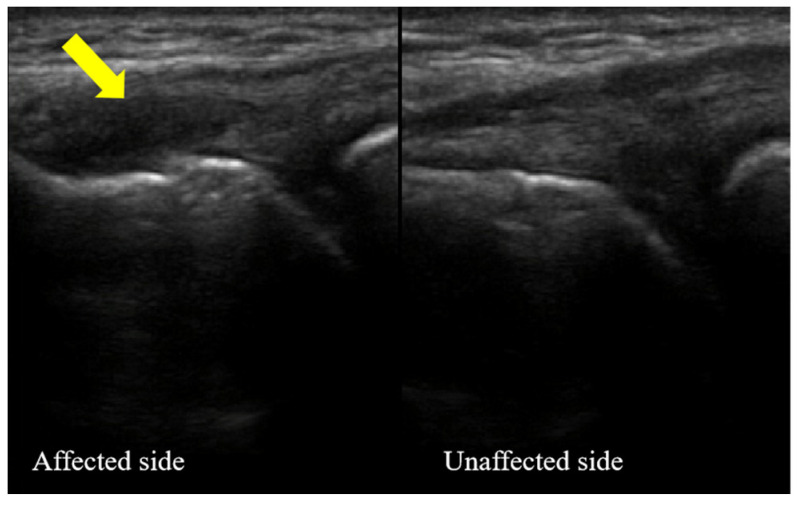
The ultrasound images of the lateral side of the left elbow affected with refractory lateral epicondylitis (case one) at the baseline. Bilateral comparison of the common extensor tendons: left elbow (affected side) and right elbow (unaffected side). The arrow points to the diseased tendon.

**Figure 2 jpm-13-00066-f002:**
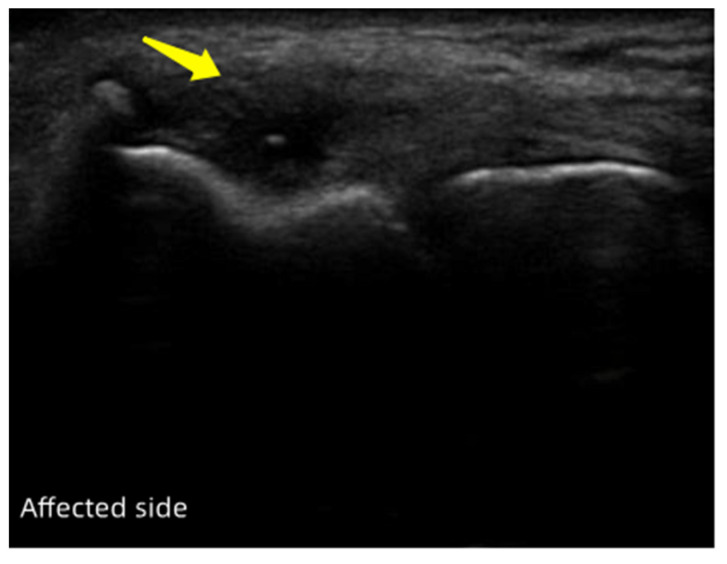
The ultrasound images of the lateral side of the right elbow affected with refractory lateral epicondylitis (case two) at the baseline. The arrow points to the damaged tendon.

**Figure 3 jpm-13-00066-f003:**
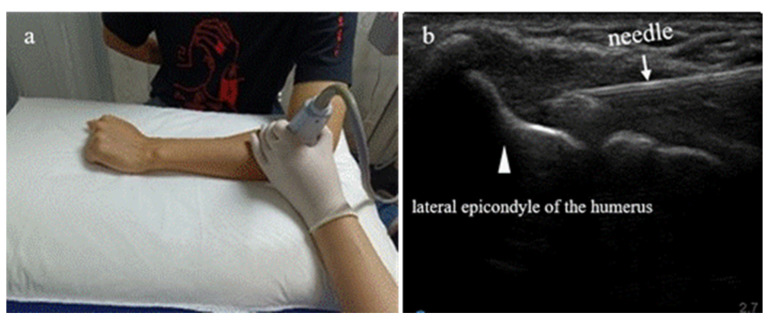
Summary of ultrasound-guided platelet-rich plasma (PRP) injection. Patient with refractory lateral epicondylitis sitting in front of a table with his forearm on it. (**a**) Ultrasound images of the radial head, the lateral epicondyle of the humerus, and the surrounding soft tissues obtained with a 13–6 Mhz probe. (**b**) Ultrasound-guided PRP administration into the injured common tendon of the external epicondyle of the humerus.

**Figure 4 jpm-13-00066-f004:**
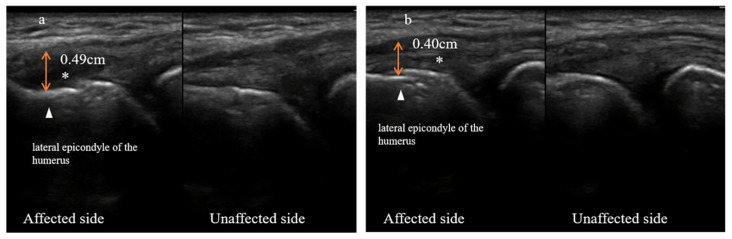
Ultrasound images of the lateral side of the left elbow of a patient with refractory lateral epicondylitis (case one) at baseline (**a**) and four weeks from the baseline (**b**). Bilateral comparison of the common extensor tendons: left elbow (affected side) and right elbow (unaffected side). The triangle points to the lateral epicondyle of the humerus. The asterisk marks the anechoic area formed by the extensor tendon tear. Double-arrowhead solid lines show the thickening of the extensor tendons.

**Figure 5 jpm-13-00066-f005:**
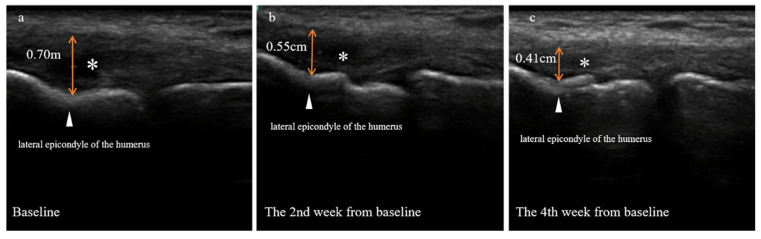
Ultrasound images of the lateral side of the right elbow of a patient with refractory lateral epicondylitis (case two) at baseline (**a**), and two (**b**) or four (**c**) weeks from baseline. The triangle refers to the lateral epicondyle of the humerus. The asterisk denotes the anechoic area formed by the torn extensor tendon. Double-arrowhead solid lines show the thickening of the extensor tendons.

**Table 1 jpm-13-00066-t001:** Visual analogue scale (VAS) and patient-rated tennis elbow evaluation (PRTEE) scores of a patient with refractory lateral epicondylitis (case one) during baseline (before the first PRP injection) and four weeks or three months after baseline.

Changes in VAS Score and PRTEE Score of Case One					
	VAS (at Rest) (0–10)	VAS (during Activity) (0–10)	PRTEE (0–100)	Pain Subscale of PRTEE (0–50)	Functional Subscale of PRTEE (0–50)
Baseline (before the first PRP injection)	4	7	53.5	27	26.5
4 weeks after baseline (Day 28)	2	3	15.5	9	6.5
3 months after baseline (Day 90)	0	2	6.5	4	2.5

**Table 2 jpm-13-00066-t002:** Visual analogue scale (VAS) and patient-rated tennis elbow evaluation (PRTEE) scores of a patient with refractory lateral epicondylitis (case two) during baseline (before the first PRP injection) and four weeks or three months after baseline.

Changes in VAS Score and PRTEE Score of Case Two					
	VAS (at Rest) (0–10)	VAS (during Activity) (0–10)	PRTEE (0–100)	Pain Subscale of PRTEE (0–50)	Functional Subscale of PRTEE (0–50)
Baseline (before the first PRP injection)	4	8	65.5	30	30.5
4 weeks after baseline (Day 28)	2	4	18	10	8
3 months after baseline (Day 90)	0	2	9	5	4

## Data Availability

Guohang, Huang (2022): PRTEE data of the case report.docx. figshare. Dataset. https://doi.org/10.6084/m9.figshare.21300792.v2.

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
