# Peer review of "Ultrasound-Guided Injection of Autologous Platelet-Rich Plasma for Refractory Lateral Epicondylitis of Humerus: Case Series"

_jpm, 2022, doi:10.3390/jpm13010066_

Round 1

Reviewer 1 Report

An interesting case study that will be of great importance in the treatment of RLE. However, I have a few comments.

1. Functional subscale should be divided by 2 [(specific activities + usual activities)/2] which adds up with the pain subscale score to give the PRTEE. The same applies to Table 1. Please correct accordingly.

2. Please include the full meaning of PRTEE at the first mention.

Line 231: is it 2nd or 4th week? Please correct accordingly

Author Response

请参阅附件。

Reviewer 2 Report

The case report series entitled “Ultrasound-guided injection of autologous platelet-rich plasma for refractory lateral epicondylitis of humerus”, has been well written. There are several questions and comments related to this manuscript, namely:

1. What is the therapeutic strategy with an ultrasound-guided injection of autologous platelet-rich plasma for refractory lateral epicondylitis at the home where the patient is being treated? Is this approach very prospective to be applied as a standard therapy?

2. All parts of the manuscript require spaces between sentences ending with citation numbers.

3. On page 2, paragraph 2 (lines 57-61) can be omitted

4. In the diagnostic assessment, the author(s) have written a score of 0-10 regarding pain level. In case 1, it says VAS 4cm/10cm (lines 92-93). It must be written consistently without units. Likewise, on line 119 for case 2.

5. The results of the pain assessment must also be written down consistently with the scores from the VAS lines 199-200 and lines 213-214, and table 1. Meanwhile, changes can be written in percentage form.

6. The table's heading must be written above the table (page 6, lines 224-225).

Author Response

请参阅附件。

Reviewer 3 Report

I would like to thank the authors for this manuscript. Interesting and concise. There are several major changes to be implemented before publication.

General issues

There are several orthographic mistakes, but the extent of the grammatical ones is too severe to neglect. I am not referring to language proficiency, these mistakes cause inconsistencies across the whole manuscript. The meaning of the sentences is guessed.  Some cause-effects logics become dubious. This can be detected in the first line of the Abstract, Introduction…etc. A sentence example full of mistakes:

 Lateral epicondylitis (LE) also known as tennis elbow, is a tendinopathy that caused by acute or chronic injuries, and the typical presenting symptoms include pain with prolonged wrist extension activities, pain with resisted wrist or elbow extension, and pain at rest radiating from the elbow along the dorsum of the forearm.

The good news is the text is short, considering that the whole manuscript needs to be severely revised. It is mandatory. Nonetheless, there are a few other basic mistakes. Check double spaces. Check spaces before “(“ and similar. All acronyms need to be defined the first time they are mentioned (for example VAS score).

Abstract + Introduction

The statement “Even though they have been received local glucocorticoid, the pain recurred soon”....

Please state the dose and timelapse for pain recurrence in the abstract. The patients were taken or had taken Glucocorticoids before the RLE diagnosis? Why were taking glucocorticoid? How much time had passed between the last intake and the RLE diagnosis?

The statement “Despite its relatively high prevalence, there is no single effective and consistent algorithm of management”

The use of algorithms here is interesting. Are you referring to an actual mathematical predictive algorithm? A medical guideline? Or it is metaphorical?

Every statement needs a citation. This includes a citation for RLE high prevalence, no effective management, use of oral NSAIDs…etc.

Considering the PrP controversy. State at least one statement supported by literature against PrP use is needed.

Material & Methods

Case descriptions:

Case 1

Information regarding Steroid dose and the number of interventions for 2016-2020 period is missing.

Case 2:

Please, add more information regarding “hormone injection” treatment.

Results

Please separate Table 1. Table information should be just after Case 1 and Case 2 Pain and functional outcome.

Ultrasound evaluation

Please improve and extend this critical section. Statements should be clearer and more precise. The results section is not for data interpretation nor to reach conclusions. It is to state simple data facts: qualitative and quantitative.

It is not clear where the Figure legend starts and ends. It is widely accepted that figures + legends must be stand-alone readable and understood. Again, state facts. In the main text should be all the facts. Facts cannot be placed in the figure legend and not in the main text.

It would be better if both cases presented the same ultrasound images. Baseline, 2 weeks, 4 weeks for affected and non-affected sides. Even better would be to have a 3-month imaging.

Discussion

*Major problems in grammar, text cohesion, causality…etc in the whole section. The first line “RLE due to its complicated pathogenesis and structural changes, patients with poor 265 curative effect on conservative treatment” is incorrect. Moreover, as a matter of writing style do not start a sentence with “9%”.The text structure and ideas pipeline is fine.

This manuscript is about the use of ultrasonography to detect structural abnormalities in PrP

Considering the PrP controversy. There should be an equilibrium between statements and citations for and against PrP. You must extend the literature against it. Furthermore, the phrase “As a biological therapy that promotes tissue repair, “ cannot be accepted for the same reasons. It implies a use, a conclusion that the medical and scientific community haven’t reached.  You may write “As a biological therapy that may promote tissue repair,” as an example.

Check language precision, and accuracy, and avoid misleading terms. For example, “Several studies have shown that RLE patients treated with PRP are more satisfied with upper limb function, “. The term “more satisfied” cannot be accepted.

At the end of “Although a meta-analysis by de Vos shows that PRP has no significant improvement effect on chronic lateral epicondylitis compared with other treatments[18]“ paragraph and before proceeding to the last two paragraphs, a few statements are missing. First, the author should address ¿why do exist these differences in PrP treatment efficacy? Different PrP methodology, different cohorts of patients, intrinsic differences among patients…etc.

The last paragraphs address the use of ultrasound evaluation. Is there any other alternative? What it is the standard procedure?

The conclusions are perfect.

Author Response

请参阅附件。
